# Increased Work from Home and Low Back Pain among Japanese Desk Workers during the Coronavirus Disease 2019 Pandemic: A Cross-Sectional Study

**DOI:** 10.3390/ijerph182312363

**Published:** 2021-11-24

**Authors:** Akira Minoura, Tomohiro Ishimaru, Akatsuki Kokaze, Takahiro Tabuchi

**Affiliations:** 1Department of Hygiene, Public Health and Preventive Medicine, Showa University School of Medicine, Tokyo 142-8555, Japan; akokaze@med.showa-u.ac.jp; 2Department of Environmental Epidemiology, Institute of Industrial Ecological Sciences, University of Occupational and Environmental Health, Kitakyushu 807-8555, Japan; ishimaru@med.uoeh-u.ac.jp; 3Cancer Control Center, Osaka International Cancer Institute, Osaka 541-8567, Japan; tabuti-ta@mc.pref.osaka.jp

**Keywords:** COVID-19, work from home, low back pain, Japan, desk worker, cross-sectional study, Internet survey

## Abstract

To prevent the spread of coronavirus disease 2019 (COVID-19), desk workers in Japan have been encouraged to work from home. Due to rapidly increased working from home, working in environments that are not properly designed and working with poor posture can affect low back pain (LBP). This study aimed to examine the relationship between increased work from home during the COVID-19 pandemic and LBP among Japanese desk workers. Using study data from the Japan COVID-19 and Society Internet Survey 2020 conducted from August to September 2020, 4227 desk workers who did not have LBP before the COVID-19 pandemic were analyzed out of 25,482 total respondents. Odds ratios (ORs) and 95% confidence intervals (CIs) for LBP were calculated by multivariable logistic regression, adjusting for covariates such as socioeconomic factors. During the COVID-19 pandemic, 31.3% of desk workers with an increased chance of working from home, and 4.1% had LBP. Desk workers with increased working from home were more likely to have LBP (OR: 2.00 (95% CI, 1.36–2.93)). In this large-scale study, increased work from home was associated with LBP among desk workers during the COVID-19 pandemic. Therefore, preparing an appropriate work environment for desk workers working from home can improve productivity, leading to positive effects.

## 1. Introduction

Coronavirus disease 2019 (COVID-19) is a global public health emergency that continues to spread around the world [1]. On 7 April 2020, the Japanese government declared a state of emergency because of the COVID-19 pandemic [2]. Japanese policies for the COVID-19 pandemic included the government temporarily closing certain businesses and “requesting” that the public refrain from going out except for emergencies. This policy led most companies to adopt remote working rapidly for business continuity as well as infection control. As the state of emergency related to the COVID-19 pandemic was extended nationwide, the proportion of companies adopting remote work rose from 20% to 60% [3].

Without a suitable work environment, working from home may have negative impacts on the health of workers. Previous studies have shown that working from home may be associated with low back pain (LBP) [4,5]. LBP is one of the leading causes of reduced healthy life expectancy, so it is important to have a better understanding of risk factors for occupational health [6,7,8,9]. The high incidence of LBP among desk workers is caused by prolonged sitting times and poor body posture, as well as other environmental factors [10]. It is possible that workers were not able to prepare their work environments appropriately for remote working because of the unexpected rapid spread of COVID-19. While the impact of working from home during the COVID-19 pandemic on LBP is not fully understood, we hypothesized that increased work from home is a risk factor for LBP. The prevalence of LBP among desk workers is also expected to have negative impacts on business management, such as a decreased labor force and increased medical costs.

However, the negative effects of increased work from home due to the COVID-19 pandemic on LBP remain unclear. Therefore, the aim of this study was to examine the relationship between increased work from home and LBP among Japanese desk workers.

## 2. Materials and Methods

### 2.1. Study Design and Setting

The Japan COVID-19 and Society Internet Survey (JACSIS) took an epidemiological approach to investigate the social and health situations of individuals in relation to the COVID-19 pandemic (https://takahiro-tabuchi.net/jacsis/howtouse/ (accessed on 11 November 2021)). In the JACSIS, 28,000 respondents were selected from 224,389 qualified panelists selected from among the approximately 2.2 million panelists registered with a Japanese Internet survey agency (Rakuten Insight, Inc., Tokyo, Japan). The participants were recruited using a random sampling method to select a sample representative of the official demographic composition of Japan as of 1 October 2019, based on each stratum by the combinations of age, gender, and region of residence (i.e., all 47 prefectures). This study employed a cross-sectional design intended to detect changes in individual lifestyle and social factors before and after the COVID-19 pandemic.

### 2.2. Study Population

We analyzed data from 25,482 participants (12,809 women and 12,673 men) after excluding 2518 participants who provided invalid responses. Measures to validate the data quality consistently were performed as described in previous studies [11,12,13,14]. Invalid responses were detected using a dummy item, “Please choose the second option from the bottom”, and 1955 participants who chose an option other than the indicated one were excluded. Next, 422 participants who answered that they used all of the listed 9 recreational substances and medications (e.g., sleeping pills, anxiolytic agents, legal/illegal opioids, cannabis, cocaine) were excluded, as were 141 participants who answered “yes” to having all of the listed 16 chronic diseases (e.g., diabetes, asthma, stroke, ischemic heart disease, cancer, mental disease). A total of 17,984 non-desk workers were also excluded, along with 2439 desk workers who had LBP before the COVID-19 pandemic. Furthermore, previous studies of occupational health have recommended that adolescents (age under 18 years) and older adults (age over 60 years) be excluded due to their significantly different working environment and treatment compared to other generations in Japan [15,16]. Seven adolescents (age 15–18 years) and 825 older adults (age 60–79 years) were excluded (Figure 1). Finally, 4227 desk workers were included in the analysis (Table 1). To determine whether the desk workers were accustomed to working from home before the COVID-19 pandemic, we also included a question about when they started working from home. To examine the impact of the declaration of the state of emergency on workers, all 47 prefectures of Japan were grouped into the following three areas based on the duration of the state of emergency declaration: “long” included the seven prefectures with the longest emergency declarations, “medium” included the six prefectures with the second longest emergency declarations, and “short” included the 34 prefectures with the shortest emergency declarations.

### 2.3. Exposure Variables

The exposure variable was increased work from home due to the COVID-19 pandemic. To examine the impact of working from home, we categorized “yes” and “no/not applicable” regarding increased work from home compared to before the COVID-19. Desk workers who answered “no/not applicable” may include without working from home.

### 2.4. Outcome Variable

LBP was assessed by the following single question: “Have you had low back strain/pain within the last month”. The responses were categorized as follows: “No”, “Yes, it developed before the COVID-19 pandemic”, or “Yes, it developed during the COVID-19 pandemic”. A similar question was used in a previous study during the COVID-19 pandemic [17]. Participants who responded “Yes, it developed before the COVID-19 pandemic” were excluded from the present study because the purpose was to examine LBP caused by the COVID-19 pandemic only. Based on the responses, the participants were classified into one of the following two categories: “LBP^−–^” (without pain) and “LBP^+^” (pain beginning during the COVID-19 pandemic).

### 2.5. Adjustment Variables

This study adjusted for participants’ demographic characteristics and labor-related variables. The participants’ demographic characteristics included age (categorized as 15–29, 30–44, and 45–59 years), gender [18], overweight and obesity (defined as body mass index [BMI] > 25) [19], educational attainment (categorized as “high school or lower” or “college or higher”) [18], marital status (categorized as “married”, “never married”, or “widowed or separated”) [20], having children [21], and outdoor exercise during the COVID-19 pandemic (“yes” or ”no”) [19]. Labor-related variables were employment status (“employer”, “self-employed”, “regular employee”, and “non-regular employee”) [22], income level (low: <4 million JPY; intermediate: 4–7 million JPY; high: >7 million JPY; and unknown or no answer) [20], and working time (<35, 35–39, 40–44, or >45 h/week) [23]. We adjusted for health-related status as follows: smoking status (never, former, and current), alcohol use (never, former, and current), and comorbidities (hypertension, diabetes, asthma, chronic obstructive pulmonary disease, cardiovascular disease, stroke, cancer, and psychiatric disorders) [24,25,26].

### 2.6. Statistical Analyses

Multivariable logistic regression analysis was applied to evaluate the association between individual working styles during the COVID-19 pandemic and LBP among Japanese desk workers. Table 2 shows the association between increased work from home and LBP during the COVID-19 pandemic. Model 1 was adjusted for age and sex. Model 2 was adjusted for exposure and adjustment variables (overweight or obesity, educational attainment, marital status, having children, outdoor exercise during the COVID-19 pandemic, smoking status, alcohol use, comorbidities, employment, income level, and work time), in addition to the adjustment factors in Model 1.

Adjusted percentages are shown with 95% confidence intervals (CIs) that were calculated using the Wald and exact methods. All statistical analyses were performed using JMP 16.0 (SAS Institute Inc. Cary, NC, USA). This study followed the STROBE guidelines for cross-sectional studies.

## 3. Results

Table 1 summarizes the characteristics of the study participants. The prevalence of LBP during the COVID-19 pandemic was 4.1% among desk workers. During the COVID-19 pandemic, 31.3% of desk workers experienced increased work from home. About 10% of desk workers had worked from home before the COVID-19 pandemic. In terms of employment, 5.6% were employers, 5.9% were self-employed, 72.7% were regular employees, and 15.8% were non-regular employees.

Table 2 summarizes the odds ratios (ORs) of increased work from home for LBP during the COVID-19 pandemic. In model 1, a significantly higher prevalence of LBP with increased work from home was observed after adjusting for age and sex (OR [95% CI]: 2.12 [1.56–2.88]). In model 2, a significantly higher prevalence of LBP with increased work from home was found (2.13 [1.52–2.97]). A significantly higher prevalence of LBP was seen among those with children (1.71 [1.10–2.66]), those with short working times (1.79 [1.04–3.08]), former smokers (1.72 [1.15–2.56]), those with cancer (3.45 [1.25–9.50]), and those with psychiatric disorders (2.33 [1.36–4.01]).

## 4. Discussion

The results of the present study revealed that increased work from home was associated with LBP among desk workers during the COVID-19 pandemic. To our knowledge, this is the first study to report an association between increased work from home and LBP during the COVID-19 pandemic.

In a previous study [10], a high incidence of LBP among desk workers was thought to be caused by prolonged sitting time and poor body posture, as well as other environmental factors in the workplace. In the present study, we did not find a significant association between the prevalence of LBP and the duration of the COVID-19 state of emergency among desk workers who reported increased work from home. We found that effects of the work environment may explain the association between working from home and LBP. Working from home has been reported to have positive effects on health (such as increasing workers’ concentration) [27]; however, it has also been reported to have negative effects on the musculoskeletal system [28,29,30]. Some studies have reported the worsening of stiff shoulders and LBP due to working from home as a result of poor posture and prolonged sitting [4]; therefore, advance guidance may be effective for new remote desk workers.

While the prevalence of LBP was higher in female than in male workers in a previous systematic review and meta-analysis [31], female were marginally associated with LBP in this study. Ikeda et al. [32] reported a significant association between the unemployment rate and LBP in the Japanese working population, especially among women. Because of the prolonged COVID-19 pandemic, it will be necessary to conduct a longitudinal study to investigate not only individual, but also social effects, including sex differences. It has been suggested that the prevalence of LBP among desk workers can lead to decreased productivity from home, but this can be avoided by preparing the home work environment appropriately [33]. A study of Chinese desk workers suggested that not having a computer monitor in front of the worker’s body and feeling cold because of low room temperature may lead to LBP [34]. Previous studies among desk workers have also reported that moderate levels of daily physical activity should be recommended for prevention of LBP [35,36,37]. Even among desk workers with increased work from home, daily moderate exercise during breaks or after work may help prevent LBP. When workers begin working from home for the first time, managers should account for these individual items as much as possible, in order to prevent LBP.

The main strength of this study is that various individual-level factors were assessed in a large-scale sample covering all 47 prefectures in Japan during the COVID-19 pandemic. On the other hand, this study has several limitations. First, we were unable to clarify the direction of causality because of the cross-sectional design. Second, the data analyzed in this study were obtained from an Internet survey and the response rate was relatively low (12.5%). However, we made as many adjustments as possible to account for possible biases in the collected sample using an external, nationally representative sample. Third, the results need to be careful about the selection bias because this study was conducted on panelists who voluntarily participated with an Internet survey company. In addition, the participants may have more time, more financial motivation, or more tolerance of a sedentary lifestyle compared with general population. This may attenuate the association between working from home and LBP. Finally, there is no information of the medical LBP diagnosis and the quality of work environment, so we could not measure pain intensity or agency and the effect of working from home accurately, respectively. Due to the COVID-19-related social distancing requirements, reduced mobility, and meeting restrictions, it was not possible to include certain measurements which could have provided relevant information regarding LBP risk factors, as observed in previous studies [38,39]. Since the increased work from home due to the COVID-19 pandemic began around March 2020, there may have been a time lag of up to 5 months until the occurrence of LBP [40]. However, although it has been difficult to treat and observe patients with LBP during the COVID-19 pandemic, we were able to collect valuable data on the status of LBP during a pandemic through this study.

## 5. Conclusions

In this large-scale study of desk workers in Japan, we found that increased work from home was associated with LBP. Moreover, female desk workers were marginally associated with LBP, and it suggested previous studies of work from home. Our results suggested that appropriately equipping the work environment of desk workers who start working from home can lead to improved productivity and well-being of workers.

## Figures and Tables

**Figure 1 ijerph-18-12363-f001:**
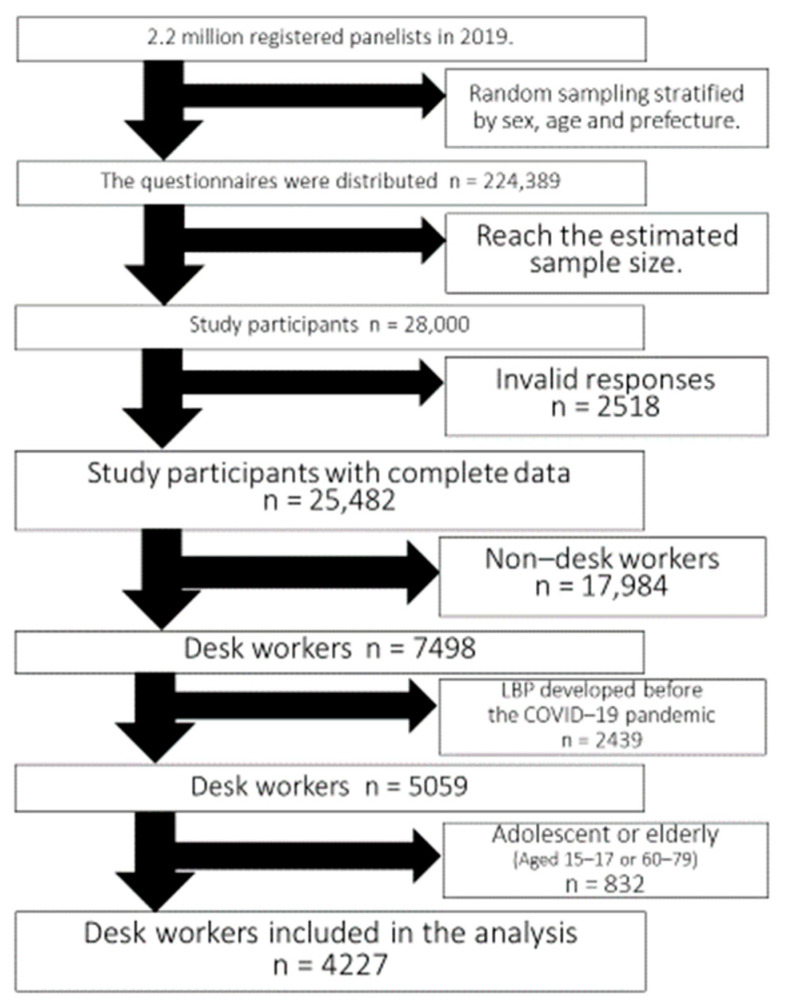
Flow diagram of the study participants.

**Table 1 ijerph-18-12363-t001:** Characteristics of study participants (desk workers aged 18–59 years).

		Total ^1^	LBP^− 2^	LBP^+ 2^	*p* Value
		(n = 4227)	(*n* = 4055 [95.9%])	(*n* = 172 [4.1%])	
Working from home has increased	Yes	1323 (31.3)	1240 (93.7)	83 (6.3)	<0.001
	No	2904 (68.7)	2815 (96.9)	89 (3.1)	
Age (years)	18–29	742 (17.5)	713 (96.1)	29 (3.9)	0.221
	30–44	1681 (39.8)	1602 (95.3)	79 (4.7)	
	45–59	1804 (42.7)	1740 (96.5)	64 (3.5)	
Sex	Female	1668 (39.5)	1602 (96.0)	66 (4.0)	0.766
	Male	2559 (60.5)	2453 (95.9)	106 (4.1)	
Overweight or obese ^3^	Yes	779 (18.4)	745 (95.6)	34 (4.4)	0.644
	No	3448 (81.6)	3310 (96.0)	138 (4.0)	
Educational attainment	College or higher	3385 (80.1)	3241 (95.7)	144 (4.3)	0.222
	High school or lower	842 (19.9)	814 (96.7)	28 (3.3)	
Marital status	Never married	1609 (38.1)	1549 (96.3)	60 (3.7)	0.599
	Married	2348 (55.5)	2246 (95.7)	102 (4.3)	
	Widowed or separated	270 (6.4)	260 (96.3)	10 (3.7)	
Have children	Yes	1703 (40.3)	1616 (94.9)	87 (5.1)	0.005
	No	2524 (59.7)	2439 (96.6)	85 (3.4)	
Outdoor exercise during the	Yes	1735 (41.1)	1645 (94.8)	90 (5.2)	0.002
COVID-19 pandemic	No	2492 (58.9)	2410 (96.7)	82 (3.3)	
Duration of the COVID-19 state of	Long	2212 (52.3)	2116 (95.7)	96 (4.3)	0.608
emergency in Japan ^4^	Medium	651 (15.4)	625 (96.0)	26 (4.0)	
	Short	1364 (32.3)	1314 (96.3)	50 (3.7)	
Employment status	Employer	236 (5.6)	221 (93.6)	15 (6.4)	0.173
	Self-employed	251 (5.9)	239 (95.2)	12 (4.8)	
	Regular employee	3071 (72.7)	2947 (96.0)	124 (4.0)	
	Non-regular employee	669 (15.8)	648 (96.9)	21 (3.1)	
Income level	Low	805 (19.0)	763 (94.8)	42 (5.2)	0.091
	Intermediate	1201 (28.4)	1150 (95.7)	51 (4.2)	
	High	1598 (37.8)	1535 (96.1)	63 (3.9)	
	Not answered	623 (14.7)	607 (97.4)	16 (2.6)	
Working time (hours per week)	<35	979 (23.2)	922 (94.2)	57 (5.8)	0.002
	35–39	633 (15.0)	612 (96.7)	21 (3.3)	
	40–44	1431 (33.8)	1391 (97.2)	40 (2.8)	
	>45	1184 (28.0)	1130 (95.4)	54 (4.6)	
Smoking status	Never	2361 (55.9)	2289 (97.0)	72 (3.0)	<0.001
	Former	900 (21.3)	847 (94.1)	53 (5.9)	
	Current	966 (22.8)	919 (95.1)	47 (4.9)	
Alcohol use	Never	782 (18.5)	754 (96.4)	28 (3.6)	0.697
	Former	1338 (31.7)	1280 (95.7)	58 (4.3)	
	Current	2107 (49.8)	2021 (95.9)	86 (4.1)	
Comorbidities	Hypertension	420 (9.9)	391 (93.1)	29 (6.9)	0.002
	Diabetes	150 (3.5)	134 (89.3)	16 (10.7)	<0.001
	Asthma or COPD	138 (3.3)	121 (87.7)	17 (12.3)	<0.001
	Cardiovascular disease	52 (1.2)	42 (80.8)	10 (19.2)	<0.001
	Stroke	31 (0.7)	25 (80.7)	6 (19.3)	<0.001
	Cancer	48 (1.1)	38 (79.2)	10 (20.8)	<0.001
	Psychiatric disorders	212 (5.0)	187 (88.2)	25 (11.8)	<0.001

LBP: low back pain; COVID-19: coronavirus disease 2019; COPD: chronic obstructive pulmonary disease. ^1^
*n* (%): % indicates the percentage of each answer. ^2^
*n* (%): % indicates the percentage of LBP^+^ or LBP^−^. ^3^ Defined as BMI > 25. ^4^ Divided into three categories as follows. Long: seven prefectures with the longest emergency declarations due to COVID-19; medium: six prefectures with the second longest emergency declarations due to COVID-19; and short: thirty-four prefectures with the shortest emergency declarations due to COVID-19.

**Table 2 ijerph-18-12363-t002:** Association between increased work from home and LBP during the COVID-19 pandemic among the desk workers.

		Model 1	Model 2
		OR (95%CI)	OR (95%CI)
Working from home has increased	Yes	2.12 (1.56–2.88)	2.13 (1.52–2.97)
	No	Ref.	Ref.
Age (years)	18–29	1.06 (0.68–1.66)	1.04 (0.61–1.77)
	30–44	1.33 (0.95–1.86)	1.28 (0.89–1.843)
	45–59	Ref.	Ref.
Sex	Female	1.00 (0.73–1.38)	1.38 (0.95–2.01)
	Male	Ref.	Ref.
Overweight or obese ^1^	Yes		0.97 (0.64–1.47)
	No		Ref.
Educational attainment	College or higher		1.23 (0.79–1.90)
	High school or lower		Ref.
Marital status	Never married		0.95 (0.57–1.57)
	Married		Ref.
	Widowed or separated		0.78 (0.36–1.66)
Have children	Yes		1.71 (1.10–2.66)
	No		Ref.
Outdoor exercise during the COVID-19	Yes		1.32 (0.95–1.82)
pandemic	No		Ref.
Duration of the COVID-19 state of	Long		1.20 (0.82–1.74)
emergency in Japan ^2^	Medium		1.15 (0.70–1.90)
	Short		Ref.
Employment status	Employer		1.31 (0.71–2.40)
	Self-employed		0.97 (0.50–1.88)
	Regular employee		Ref.
	Non-regular employee		0.56 (0.32–0.98)
Income level	Low		1.42 (0.89–2.25)
	Intermediate		Ref.
	High		0.75 (0.50–1.13)
	Not answered		0.70 (0.39–1.26)
Working time (hours per week)	<35		1.79 (1.04–3.08)
	35–39		Ref.
	40–44		0.82 (0.48–1.43)
	>45		1.30 (0.76–2.20)
Smoking status	Never		Ref.
	Former		1.72 (1.15–2.56)
	Current		1.46 (0.97–2.19)
Alcohol use	Never		Ref.
	Former		0.97 (0.60–1.58)
	Current		0.95 (0.59–1.52)
Comorbidities	Hypertension		1.43 (0.87–2.36)
	Diabetes		1.45 (0.72–2.92)
	Asthma or COPD		1.26 (0.61–2.61)
	Cardiovascular disease		1.44 (0.45–4.58)
	Stroke		0.37 (0.08–1.72)
	Cancer		3.45 (1.25–9.50)
	Psychiatric disorders		2.33 (1.36–4.01)

LBP, low back pain; COVID-19, coronavirus disease 2019; COPD, chronic obstructive pulmonary disease; OR, odds ratio; CI, confidence interval. ^1^ Defined as BMI > 25. ^2^ Divided into three categories as follows. Long: seven prefectures with the longest emergency declarations due to COVID-19; medium: six prefectures with the second longest emergency declarations due to COVID-19; and short: thirty-four prefectures with the shortest emergency declarations due to COVID-19. Model 1: adjusted for age and sex. Model 2: in addition to Model 1, adjusted for overweight or obesity, educational attainment, marital status, having children, outdoor exercise during the COVID-19 pandemic, psychological distress, smoking status, alcohol use, comorbidities, employment, income level, and working time. Pseudo-R^2^ statistics R^2^ (Model 1) = 0.018 R^2^ (Model 2) = 0.082.

## Data Availability

The data sets used in this study are not available in a public repository but are available from the corresponding author upon reasonable request.

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
