# Peer review of "Increased Work from Home and Low Back Pain among Japanese Desk Workers during the Coronavirus Disease 2019 Pandemic: A Cross-Sectional Study"

_ijerph, 2021, doi:10.3390/ijerph182312363_

Round 1

Reviewer 1 Report

The study by Minoura et al. has highlighted the role of LBP due to increased work from home using survey data. the statistics and the analysis done in this paper is consistent with standards. 

Author Response

Thank you for the peer review. We checked the English in our manuscript.

Best regards.

Reviewer 2 Report

This study attempts to investigate the potential LBP-inducing effect of increased work from home during the COVID-19 pandemic. It is well-designed, and the manuscript is well-written. However, I have several concerns as described below.

Major

p. 2, line 60-64: “28,000 respondents were selected from 224,389 … as of October 1, 2019.”

This passage is unclear. If my understanding was correct, 28,000 respondents were NOT selected; the passage should read: “the investigators randomly sampled 224,389 out of 2.2 million panelists who were voluntarily registered in a Japanese Internet survey agent. The sampling was stratified by age and sex so that the demographic composition of the sample should represent that of the Japanese people as of October 1, 2019. Out of the 224,389 registered panelists, 28,000 (12.5%) responded the questionnaire.”

In my view, it is also important to described how 2.2 million panelists were recruited and registered (was the participation completely voluntary; was the recruitment based on demographic characteristics?). This is relevant to generalizability.

In addition, was the representativeness of the demographic composition realized among the respondents? This is information relevant to potential selection bias.

Furthermore, under what title, was the participation requested to the potential 224,389 participants? If that said, “the survey on LBP and working from home,” for example, it might have attracted panelists who developed LBP during working from home, a preferential participation that could introduce selection bias; in that case, the association could be overestimated. Alternatively, if the presence of LBP or that of increased working from home increased response independently, it could underestimate the association because of collider bias.

Figure 1:

It should start from the 2.2 million registered panelists.

p. 4, line 115: “Have you had LBP or fatigue …”

This question asks the development of LBP or fatigue, not LBP alone. This is the most serious of my concerns in the study. Do you have any rationales to justify equating “LBP or fatigue” with LBP alone? Otherwise, this study is critically flawed at the start.

Minor

p. 3, line 77:

I could not fully understand why those 141 participants were “invalid” who answered “yes” to having one or more of the listed chronic diseases. Please explain the rationales. Also, table 1 shows some of the “valid” participants have the chronic disease listed. It seems contradictory.

p. 3, line 81:

Why were the adolescents and older adults excluded? Also, if the exclusion justified, why did you not do so at the sampling stage?

Table 1.

To help compare the covariate distributions, the percentages should be those in the subpopulations, i.e., in the columns “LBP-“ and “LBP+,” the numerators should be 4,055 and 172, respectively.

p. 4, line 99: Exposure variables

If I understood correctly, in light of the study’s aim, the only exposure variable is increased work from home; all the others should be adjustment variables.

p. 7, line 177-178: “Our results indicated that a short duration of working from home may be associated with LBP among desk workers.”

If my understanding was correct, the investigators did not ask the duration of working from home; they asked about “working time,” which included working time both form home and at office. Consequently, the authors’ inference seems weak.

p. 7, line 186-188: “this study did not find any gender differences in the prevalence …”
In model 2, the OR of female is 1.38 (0.95-2.01). Although it did not reach statistical significance, it may suggest female predominance.

p. 7, line 206-210, limitations:

The discussion should describe potential limitations inherent to Internet surveys more concretely. In Internet surveys, there are four occasions of introducing selection bias: 1) during recruiting panelists, 2) whether the potential panelists are willing to registered in the panel, 3) during recruiting the participants in a particular survey out of the registered panelists, and 4) whether the potential participants are willing to respond. The first three are concerned with external validity (generalizability), and the last with internal validity (validity of inference). The low response rate of 12.5% raises a concern with the last possibility (internal validity) because the response may be biased, possibly related to outcome or unmeasured confounders—for example, a possibility I mentioned earlier. The generalizability could be jeopardized not only by access/literacy to the Internet, but by such factors as financial motivations, available time, etc. The latter factors could also affect response to this particular survey (including whether the response became invalid), hence a potential hazard to the internal validity. I appreciate the investigators’ efforts, such as stratified random sampling, dummy question, and comprehensive covariate collection, to secure validity as much as possible. Such measures might be mentioned as needed.

Author Response

We would like to thank the editor for the feedback that helped us improve the manuscript. Our detailed point-by-point responses to each of the comments.

Reviewer 3 Report

This article seeks to identify whether there is a relationship between increased work from home during the COVID 19 pandemic and low back pain among desk workers. To do so, the authors use a logistic regression.

The following comments are offered as constructive suggestions with the goal of improving the article.

  1. Overall, I find this article to be quite confusing and some points could use some clarification. First, I don’t understand how the interest variable “increased work from home” is measured. Is it the fact to not work at home before the COVID and to work at home during this period? Or is it the fact to have increase the hours spend working at home during the pandemic? The authors should clarify this point. If the first case is the wright, authors should not use the word increase. Second, the way that the sample was presented is sometimes difficult to understand. For example, lines 77-79 authors wrote that survey participants who suffering from chronic diseases are excluded but in the Table 2, these chronic diseases are used as explanatory variables. Third, the way that the authors classified their variables between “exposure variables” and “adjustment variables” are unclear. Indeed, some variables listed in exposure variables (like educational attainment, gender, …) in section 2 are in section 3 interpreted as adjusting variables (for example line 151-152). Moreover, line 110, a reference to unemployed is done while unemployed are excluded from the analysis. Fourth, some wordings are confusing (line 22: “31.3% of desk workers had a higher chance of working from home…”) (line 14-15: is not the increasing number of remote workers that cause LBP)….
  2. Authors should provide information on the quality of their models. They should also give information on the standard deviations and on the significance or not of their variables.
  3. Line 186-188: authors should present the result link to this conclusion. The interaction term they test, should presented in a Model 3.
  4. Do authors have information on the quality of work environment (workstation ergonomics)?
  5. Do all desk workers who don’t increase their frequency of working from home worked during the pandemic? If some of them couldn’t work, it could be interesting to divide the workers who didn’t increase their frequency of work from home between workers who work and the others.
  6. The authors stress the fact that their depending variable are LBP but to measure LBP they use a question based also on fatigue (l115-116). Authors should clarify this point.

Author Response

(The authors gave the same response as above.)

Round 2

Reviewer 2 Report

I appreciate the authors responding some of my concerns. However, substantial concerns remain intact in the manuscript. Above all, the authors should fully understand the methods in JACSIS themselves, and they should describe them correctly.

  1. “In this study, data were obtained … for each category (sex, age, and prefecture)”

I do understand the sampling process, but readers must! The authors should describe that in the manuscript. In addition, I have several comments on the above description:

1) Does “each category (sex, age, and prefecture)” mean “each stratum by the combinations of sex, age, and prefecture?” (Did you aim at attaining sufficient sample sizes for each sex, age, and prefecture category or those for each stratum by their combinations?) In other words, was your aim marginal distributions or joint distributions. If the latter was the case, please rephrase it.

2) The authors should cite the website of JACSIS (https://takahiro-tabuchi.net/jacsis/howtouse/), where the investigators present the flow of the survey and the whole questionnaire distributed to the panelists. Although it is shown in only Japanese language, with the aids of translation applications, it should help anyone understand the survey methodology.

3) The authors should briefly describe the questionnaire, e.g., how many and what kind of questions it contained. It should help readers understand that this study is a sub-study of the inclusive survey.

  1. Exclusion of participants with chronic diseases.

The authors’ response (exclusion of those who have all 16 diseases) clearly conflicts with the description in the manuscript: “as were 141 participants who answered “yes” to having one or more of the listed chronic diseases (e.g., diabetes, asthma, stroke, ischemic heart disease, cancer, mental disease)” I found a reasonable description in another article from JACSIS [Int. J. Environ. Res. Public Health 2021, 18(17), 9406]: “Out of 28,000, those who gave inappropriate answers (i.e., those whom we speculated not reading the questionnaire; n = 2518) were excluded by checking whether answers were consistent with a developed algorithm [13,14]. In the algorithm, those who failed to respond to our dummy question, “Please choose the second item from the bottom of a list”, and who chose every item in the questions “Select which drugs are used”, with a list of 7 substances, and “Choose which chronic diseases apply”, with a list of 16 diseases, were excluded.” Now I understand that the exclusion was based on the putative invalidity of the responses not on having the diseases. If that was the case, please mention it.

  1. LBP or fatigue.

I looked into the corresponding question on the above website. It says in Japanese “Koshino-tsukare/itami,” translated verbatim into English, “low back fatigue/pain.” I guess it may be translated into “low back strain/pain.” The original phrase “low back pain or fatigue” is quite misleading because it seems as if the question asked if the participants had LBP or generalized fatigue.

  1. Line 226: “The results need to be careful about the bias due to self-reporting about pain.”

This sentence seems to be concerned with potential bias due to preferential self-recall (recall bias) or to voluntary participation in Internet survey (selection bias). On the other hand, the preceding sentence is concerned with accuracy of exposure and outcome variables, hence a different kind of limitation. Accordingly, this should be discussed independently or together with the third limitation. Furthermore, the authors should explicitly discuss the fact that voluntary participation in Internet survey could not only undermine generalizability but also introduce selection bias. In addition, this sentence appears grammatically peculiar. Please have another editing for English language.

  1. Line 222: “our findings may not be generalizable to populations with limited access/literacy to the Internet”

This is true, but can persons who have limited access/literacy to the Internet work from home? The limited generalization could stem more from the self-selection of the participants, where the participants may have more time, more financial motivation, or more tolerance to sedentary lifestyle compared with general population.  

Author Response

We would like to thank again for the feedback that helped us improve the manuscript. Our detailed point-by-point responses to each of the comments.

Reviewer 3 Report

The authors did not answer to point 5, which is crucial to know if their conclusions are adequate.
No information on the significance of the variables in the logistic model is given to check if there is a significant link between the dependant variable and independant variables. No information is given on the quality of the model (such as the pseudo R2).

In point 2, authors answered that only participants who answered all 16 diseases in the questionnaire as comorbidities are dropped but it's not writen like that line 78.

Author Response

(The authors gave the same response as above.)
